# Impact of Hyper- and Hypo-Uricemia on Kidney Function

**DOI:** 10.3390/biomedicines11051258

**Published:** 2023-04-24

**Authors:** Junichiro Miake, Ichiro Hisatome, Katsuyuki Tomita, Tadahiro Isoyama, Shinobu Sugihara, Masanari Kuwabara, Kazuhide Ogino, Haruaki Ninomiya

**Affiliations:** 1Division of Pharmacology, Department of Pathophysiological and Therapeutic Science, Tottori University Faculty of Medicine, Tottori 683-8503, Japan; 2Department of Cardiology, Yonago Medical Center, Tottori 683-0006, Japan; 3Department of Respiratory Disease, Yonago Medical Center, Tottori 683-0006, Japan; 4Department of Urology, Yonago Medical Center, Tottori 683-0006, Japan; 5Health Service Center, Shimane University, Matsue 690-0823, Japan; 6Intensive Care Unit and Department of Cardiology, Toranomon Hospital, Tokyo 105-8470, Japan; 7Department of Cardiology, Tottori Red Cross Hospital, Tottori 680-0017, Japan; 8Department of Biological Regulation, Tottori University Faculty of Medicine, Tottori 683-8503, Japan

**Keywords:** hyperuricemia, hypouricemia, urolithiasis, tubular disease, kidney disease, U-shaped association, endothelial function, xanthine oxidase, uric acid transporters

## Abstract

Uric acid (UA) forms monosodium urate (MSU) crystals to exert proinflammatory actions, thus causing gout arthritis, urolithiasis, kidney disease, and cardiovascular disease. UA is also one of the most potent antioxidants that suppresses oxidative stress. Hyper andhypouricemia are caused by genetic mutations or polymorphism. Hyperuricemia increases urinary UA concentration and is frequently associated with urolithiasis, which is augmented by low urinary pH. Renal hypouricemia (RHU) is associated with renal stones by increased level of urinary UA, which correlates with the impaired tubular reabsorption of UA. Hyperuricemia causes gout nephropathy, characterized by renal interstitium and tubular damage because MSU precipitates in the tubules. RHU is also frequently associated with tubular damage with elevated urinary beta2-microglobulin due to increased urinary UA concentration, which is related to impaired tubular UA reabsorption through URAT1. Hyperuricemia could induce renal arteriopathy and reduce renal blood flow, while increasing urinary albumin excretion, which is correlated with plasma xanthine oxidoreductase (XOR) activity. RHU is associated with exercise-induced kidney injury, since low levels of SUA could induce the vasoconstriction of the kidney and the enhanced urinary UA excretion could form intratubular precipitation. A U-shaped association of SUA with organ damage is observed in patients with kidney diseases related to impaired endothelial function. Under hyperuricemia, intracellular UA, MSU crystals, and XOR could reduce NO and activate several proinflammatory signals, impairing endothelial functions. Under hypouricemia, the genetic and pharmacological depletion of UA could impair the NO-dependent and independent endothelial functions, suggesting that RHU and secondary hypouricemia might be a risk factor for the loss of kidney functions. In order to protect kidney functions in hyperuricemic patients, the use of urate lowering agents could be recommended to target SUA below 6 mg/dL. In order to protect the kidney functions in RHU patients, hydration and urinary alkalization may be recommended, and in some cases an XOR inhibitor might be recommended in order to reduce oxidative stress.

## 1. Introduction to Hyperuricemia and Hypouricemia: Epidemiology and Genetic Mutation

When serum uric acid (SUA) levels exceed 6.8 mg/dL, the ionized forms of UA form monosodium urate (MSU). When the MSU concentration exceeds the solubility, it forms MSU crystals [1], which in turn causes gout arthritis and urolithiasis. Gout is characterized by joint arthritis and is frequently accompanied by urolithiasis and interstitial nephritis [2]. Epidemiological studies have demonstrated that an increase in SUA is associated with hypertension, heart failure, diabetes, and metabolic syndrome [3,4,5]. Mechanistically, UA in the blood acts on endothelial cells to reduce nitric oxide (NO) levels, and xanthine oxidase (XO) in cytosol and on plasma membranes produces superoxide and reduces NO levels [6], causing chronic kidney disease (CKD) and cardiovascular events.

UA is a potent antioxidant and can scavenge superoxide, hydroxyl radicals, and singlet oxygen. It can neutralize more than 50% of free radicals in human blood [7]. Thus, a decrease in SUA leads to a reduced antioxidant capacity, resulting in renal disease. It is well known that renal hypouricemia (RHU) is frequently associated with exercise-induced acute kidney injury (EIAKI) [8]. Several epidemiological studies have indicated that hypouricemia is associated with impaired kidney function [9,10].

While the pathophysiological mechanisms of hyperuricemia are different from RHU, both conditions lead to elevated urate levels in proximal tubules. Since the filtered urate load, calculated as [SUA level] × [GFR], is increased in hyperuricemia patients, gout is often accompanied by interstitial nephritis. RHU increases the urate concentration in the proximal tubules due to impaired tubular reabsorption, and give rise to a similar situation. In fact, it has been reported that there is a J-shaped relationship between the level of SUA and the incidence of CKD [10,11].

This review discusses the epidemiology and genetic background of hyper- and hypo-uricemia and complications such as urolithiasis, renal tubular damage, and kidney in patients with dysuricemia including hyperuricemia, gout and hypouricemia. It also discusses the U-shaped association of SUA with impaired kidney function.

Hyperuricemia is diagnosed when the concentration of SUA is >7 mg/dL based on the guidelines of the Japanese Society of Gout, Uric Acid, and Nucleic Acids. Hyperuricemia occurs in 20% of men and 5% of women in the entire Japanese population. A total of 1.25 million Japanese suffer from gout [12]. There is a gender difference in hyperuricemia and gout. Since the prevalence of hyperuricemia in women increases and is comparable to that in men after menopause, it is believed that estrogen and progesterone secretion is responsible for the gender difference. Hyperuricemia can be caused by genetic mutations or polymorphisms that increase the production of UA or decrease its excretion [12]. However, the recent increase in the number of hyperuricemic patients indicates the presence of environmental factors that influence the prevalence of hyperuricemia.

Hypouricemia is diagnosed when the SUA level is <2 mg/dL [13]. Hypouricemia was observed in 0.09~0.21% of males and 0.36~0.51% of females [14,15]. Kuwabara et al. reported that the incidence of hypouricemia in female subjects decreases with age, suggesting the gender differences with regard to hypouricemia might be attributable to the secretion of estrogen or progesterone [16]. Hypouricemia can be classified into two types: transient or persistent hypouricemia [17]. The persistent hypouricemia is found in 0.15% of outpatients [17] (Table 1). Hypouricemia is attributable to either the decreased production or the increased urinary excretion of UA, and persistent hypouricemia is due to a genetic deficiency of UA production or tubular UA reabsorption. The former is mainly due to mutations in xanthine oxidase, which is clinically characterized by xanthinuria. The latter is due to the mutations of UATs responsible for reabsorbing tubular UA [13].

Being attributable to either increased production or decreased urinary excretion, hyperuricemia and gout are categorized as the renal overload type (overproduction type and the reduced extrarenal excretion type) and the renal underexcretion type. Several genes are reported to be associated with gout, such as ABCG2, NIPAL1, FAM35A, and ALDH1B and 2 [20]. The development of hyperuricemia has been shown to be accompanied by multiple genetic factors and mutations in the uric acid transporter protein genes SLC2A9 (encoding GLUT9), SLC22A12 (encoding URAT1), SLC17A1 (encoding NPT1) and ABCG2, which were most strongly correlated with changes in serum uric acid levels [1,8,15,21]. The uric acid transporter ATP-binding cassette superfamily G member 2 (ABCG2) is expressed not only in the kidney but also in the intestine, where it transports UA to stool [22]. The single nucleotide polymorphism of ABCG2 causes reduced extrarenal excretion type hyperuricemia and gout [23]; amino acid substitution Q126X abolishes ABCG2 function completely, whereas Q141K reduces it by 25%.

Hypouricemia is attributable to the genetic impairment of UA production or reabsorption. The impairment of production is most frequently caused by mutations in xanthine oxidase (xanthinuria). Some 90% of the filtrated UA at the glomerulus is reabsorbed through UATs. Uric acid transporter 1 (URAT1) and glucose transporter 9 (GLUT9) are localized at the apical site of the proximal tubules and at their luminal site, respectively, and play an important role in UA excretion. The impairment of URAT1 activity [24] causes renal hypouricemia (RHU) type 1, and the impairment of GLUT9 activity causes RHU type 2 [15]. A genetic analysis indicates that Japanese patients with RHU frequently possess the W258X (G774A) and G269A (R90H) URAT1 mutations [8]. The 258X allele was found in three cases of five patients with hypouricemia (60%) [25], and is found in 1.1% of the total study population [26] in Koreans. The finding that the 258X allele has not been reported in other ethnic groups suggests that the origin of the RHU-harboring mutant URAT1 gene could be in East Asia [27]. We recently reported that nine cases out of 15 patients with RHU (60%) harbored the W258X (G774A) URAT1 mutation [28].

## 2. Clinical Evidence of Hyper- and Hypo-Uricemia May Cause an Adverse Effect on Kidney

A typical complication in patients with hyperuricemia and gout is ureteral calculus, which is caused by monosodium urate (MSU) crystals. The prevalence of uric acid stones among urolithiasis varies depending on the population under study. It has been reported to be as low as 8% in the USA, whereas in Israel it has been reported to be 39.5% [29]. The prevalence of uric acid stones in the general population of the USA is approximately 0.01%, and it was approximately 22% in patients with gout. Urolithiasis associated with gout patients is mainly caused by the elevation of the SUA level, urinary UA excretion, and a reduced urinary pH [2] (Figure 1). The prevalence of urolithiasis depends on the level of SUA, since SUA > 7.0 mg/dL significantly increased the risk of urolithiasis, and this risk is augmented by 50% at SUA > 12.0 mg/dL. The prevalence of urolithiasis also depends on the urinary urate excretion; urinary excretion at UUA > 700 mg/day significantly increased the prevalence of urolithiasis and UUA > 1100 mg/day augments it by 50% [30], indicating that urinary urate excretion is the key factor of urolithiasis in gout patients. It has been documented that patients with uric acid urolithiasis have a lower pH of urine [31], which might be a result of reduced ammonia excretion.

Interestingly, RHU has been reported to be accompanied by urolithiasis [8]. We reported that RHU was associated with urolithiasis and hematuria. The renal stones in patients with RHU were radiolucent, and was abolished by urinary alkalization [32], suggesting that the increased urinary level of urate is a risk for urolithiasis in RHU. It has been considered that urolithiasis in RHU might also be due to increased levels of urinary UA concentrations and reduced urinary pH values, causing urinary MSU precipitation. Hematuria in patients with RHU is thought to be due to an elevated urinary urate concentration [33]. The excess urinary excretion of UA correlated with aciduria in RHU [34] (Figure 2).

We recently reported that the prevalence of urolithiasis in RHU patients (18.2%) was higher than that in gout patients (6.8%) [28]. Because of the similarity in pH values in urine between RHU and gout patients, the mixed calculi that consists of urate and calcium oxalate is caused by the elevated urinary UA concentrations in cases with a normal urinary pH value. This is because MSU at high levels precipitates out of a solution and is suspected to result in calcium oxalate crystallization. UA excretion estimated by the uric acid/creatinine clearance ration (Cur/Ccr) in RHU patients accompanied by urolithiases were higher than those in RHU patients without urolithiasis, and 75% of these cases with urolithiasis harbored a null function of URAT1. Thus, both an elevated urinary UA concentration and a null function of URAT1 could be dominant factors influencing urolithiasis in RHU (Figure 2).

Gout nephropathy is induced by an increased urinary uric acid excretion, which accumulates in the renal tubules and interstitum [35] (Figure 1). The filtered urate load, estimated by [SUA level] × [GFR], is elevated in gout patients, and their urate concentrations in proximal tubules are increased, resulting in gout frequently being accompanied by interstitial nephritis. MSU precipitates in the renal tubules, especially in the collecting ducts, causing acute gouty nephropathy [36]. Chronic gouty nephropathy is associated with urate crystal deposits, which are characterized by the presence of urate deposition in the interstitium and tubules as well as the formation of tophus. A cross sectional study of 502 patients found that the renal medulla of patients with severe gout was diffusely hyperechoic, supporting the idea that the renal medulla of patients with gout involves the hyperechoic region [37]. Interestingly, lowering the SUA by allopurinol has reduced not only the double contour sign of the joint but also the hyperechoic region of the renal medulla, improving eGFR. Thus, the elevated urinary uric acid together with the MSU deposition in the renal medulla could cause damage to the interstitium and tubules. Urolithiasis also impairs urine flow, causing classical gout nephropathy due to nephron damage and infection (Figure 1).

It is likely that tubular reabsorption impairment in RHU elevates the urate concentration in proximal tubules and causes tubular damage (Figure 2). RHU was reported to induce an increase in urinary levels of beta 2-microglobulin in comparison to gout patients. Since the urinary beta 2-microglobulin levels are correlated with the Cur/Ccr, the increased urinary UA levels can cause the damage to the proximal convoluted tubules in RHU [28]. The RHU cases with an elevated urinary beta 2-microglobulin level harbored the URAT1 mutation, suggesting that these cases of RHU harboring URAT1 mutation cause impaired tubular function [28].

Hyperuricemia is often accompanied by chronic kidney disease (CKD), and has been reported to be a risk factor for the progression of CKD. Indeed, a recent meta-analysis reported that 24% of gout patients exhibited CKD beyond stage 3, and that hyperuricemia occurs in advanced CKD, with a prevalence of 64% with stage 3 CKD and 50% in patients with stage 4 or 5 CKD [38,39,40]. Thus, hyperuricemia could cause the progression of CKD. Uric acid is reported to have a direct effect on the structure of the vessels such as preglomerular arteriolar damage characterized by hyalinosis and wall thickening (Figure 1). UA could promote the proliferation of vascular smooth muscle cells and endothelial dysfunction due to the activation of RAS and decreased NO levels. Kohagura et al. [41] reported an association between the uric acid level and renal arteriopathy in CKD patients using a biopsy-based study. They showed that hyperuricemia was associated with a higher risk of hyalinosis and higher-grade wall thickening of the renal artery. It has been reported that the SUA level was correlated with renal blood flow in the renal medulla and with the glomerular capillary pressures. Hyperuricemia is associated with higher renal resistive index (RRI) values and lower renal volume-to-resistive index ratio (RV/RRI) values [42]. Since the RV/RRI value has been proposed as an indicator of intrarenal arteriolopathy, hyperuricemia may cause intrarenal arteriolopathy. It has been reported that while creatinine clearance and urinary albumin excretion (UAE) were similar between two groups, RRI values were increased in hyperuricemic patients, and RV/RRI values were inversely related to SUA levels. The logistic regression analysis identified serum UA as an independent predictor of decreased RV/RRI. Taken together, in CKD patients, hyperuricemia causes renal arteriopathy including glomerular tubular and interstitum damage. The plasma XOR might have harmful effects on kidney disease. UAE was correlated to the activity of plasma XOR, and an XOR inhibitor topiroxostat reduced UAE [43]. This indicates that the plasma XOR that originated from the liver could bind glomerular vessels and elevate UAE (Figure 1).

RHU poses a risk of acute kidney injury. Exercise-induced acute kidney injury (EIAKI) is a major complication of RHU. The prevalence of EIAKI has been reported to be 9.4% in hereditary RHU subjects [8]. It has been reported that hypouricemia is accompanied by an impaired kidney function [14]. Kuwabara et al. [16] reported that hypouricemic patients with a history of kidney diseases showed a normal kidney function, suggesting that kidney diseases in patients with RHU are reversible. We recently reported that a patient with EIAKI was a homozygote for the W258X (G774A) URAT1 mutation [27]. The reason why RHU could cause EIAKI remains to be elucidated. Reduced UA could weaken antioxidant actions on the capillary, causing vasoconstriction of the renal artery and EIAK. Furthermore, the precipitation of urate crystals induced by enhanced urinary urate concentration might be attributable to EIAKI. The elevation of urinary UA levels in RHU patients may cause the intratubular and/or intraurethral precipitation of UA crystals, causing EIAKI due to the activation of the NLRP3 inflammasome. Other risk factors for precipitation of urate include a low urinary pH, low urine volume, and a decrease in the extracellular fluid volume [2] (Figure 2).

While hyperuricemia and gout are associated with kidney diseases, hypouricemia is also associated with kidney injury, indicating the U-shaped association between SUA and kidney disease. A U-shaped association between SUA and loss of kidney function has also been reported. Kanda et al. [10] reported that in subjects with low serum uric acid levels (male: SUA < 5.0 mg/dL, female: SUA < 3.6 mg/dL), low serum uric acid levels were associated with a time-dependent decline of eGFR, while low and high levels of SUA were associated with the loss of kidney function. It was also demonstrated that exercise load was associated with the loss of kidney function in males with low serum uric acid levels, suggesting the involvement of EIAKI. It has also been reported that in healthy Korean volunteers [44], low levels of SUA were related to end-stage renal disease (ESRD), while elevated levels of SUA were associated with an increased risk for ESRD, suggesting an association between SUA levels and ESRD. The U-shaped association between SUA and kidney disease could be explained by impaired endothelial function induced by either hyper-uricemia or hypo-uricemia. UA reduces nitric oxide (NO) levels in endothelium cells under hyperuricemic conditions, resulting in endothelial dysfunction [1,6]. Since UA is one of the most potent antioxidants in the extracellular space, a decrease in SUA leads to reduced antioxidant capacity, resulting in impaired endothelial function [19]. The decrease in the SUA value is also associated with the hypertrophic remodeling of small blood vessels [45].

It has been reported that there is also a U-shaped relationship between the level of SUA and the incidence of cardiovascular events (CVE) [46]. Verdecchia et al. reported an increase in CVE risk with the sUA value < 4.5 mg/dL in men and <3.2 mg/dL in women [47]. Mazza et al. also showed that in aged patients with non-insulin-dependent diabetes, an SUA < 4.9 mg/dL is a risk factor for death from CVE [48]. Extremely low sUA induced by the administration of UA lowering agents has been shown to increase CVE risk [49]. Deleeu et al. reported a U-shaped relationship between SUA and the total mortality of hypertensive patients [50]. Lee et al. [51] also suggest that there is a U-shaped relationship between SUA and the total mortality of hypertensive patients in men, and the lowest risk of CVE was observed at an SUA level of 6.9 mg/dL. Taken together, a U-shaped association between SUA and loss of kidney function as well as CVE has been observed on various occasions.

## 3. Molecular Mechanism of Hype-, Hypo-, Transient Hypo-Uricemia May Cause Endothelial Dysfunction

UA infusion into the brachial artery caused impaired vasodilation of the forearm artery in response to acetylcholine in healthy volunteers [52], suggesting that hyperuricemia impairs NO-dependent vasodilation [6,53]. This impaired vasodilation is exerted by the intracellular accumulation of UA in endothelial cells, taken up via UA transporters (UATs) [1,23,54,55,56]. Intracellular UA decreases the NO level in response to acetylcholine by suppressing its production by endothelial nitric oxide synthase (eNOS), accelerating its degradation, attenuating arginine uptake and the reduction of the binding of eNOS to calmodulin [1]. These inhibitory effects of UA on NO production were cancelled by anti-oxidants, suggesting the involvement of a superoxide. Indeed, intracellular UA increases angiotensin II production as well as superoxide generation, inducing the senescence and apoptosis of endothelial cells. Hyperuricemia may also lead to microvascular injury by the activation of RAS, inhibiting eNOS and vascular smooth muscle proliferative effects (Figure 3). Hyperuricemia increases the expression of renin in glomerular cells and that of renin receptors in endothelial cells, while it decreases NOS-1 expression in the macula [57,58].

MSU crystals are known to enhance the production of inflammatory cytokines such as IL-1β or IL-18 in primed macrophages through the activation of the NLRP3 inflammasome, leading to local or systemic pro-inflammatory reactions. This causes vascular and kidney diseases. The CANTOS study [59] demonstrated that the anti- IL-1β antibody attenuated arteriosclerosis. Furthermore, soluble UA stimulates IL-1β production through the activation of the NLRP3 inflammasome, causing tubular damage. This inflammatory action of UA on tubular cells could be through the high mobility group box chromosomal protein 1 (HMGB1) that facilitates nuclear factor kappa-B (NF-*k*B) signaling [58] (Figure 3).

Xanthine oxidoreductase (XOR) plays a central role in the formation of UA as well as the superoxide. XOR is composed of xanthine dehydrogenase (XDH) and xanthine oxidase (XO). XO is converted from XDH by post-translational modification and catalyzes two steps of reactions that convert hypoxanthine to xanthine, and then xanthine to UA. These reactions generate superoxide and hydrogen peroxide [60]. The superoxide derived from XO reacts directly with NO, resulting in both reduced NO and increased peroxynitrite. When peroxynitrite oxidizes and inactivates tetrahydrobiopterin, eNOS is converted into a superoxide-generating enzyme [61]. Recently, XO that originated from liver was reported to exist in plasma [62]. It binds glycosaminoglycans on the surface of endothelial cells [63]. The superoxide derived from endothelium-bound XO reacts with NO, generates peroxynitrite, and decreases the NO-dependent production of cGMP via smooth muscle cells [64]. Plasma XO activity has been reported to be negatively correlated to FMD [65] and eGFR [66]. Taken together, the activation of plasma XO could cause endothelial dysfunction as well as the loss of kidney function [1].

It is assumed that the extreme reduction of SUA impairs endothelial function. The endothelial function is impaired in patients with RHU harboring homozygous/compound heterozygous mutations of URAT1. Sugihara et al. reported that patients with URAT1 gene mutations and severe hypouricemia (sUA < 0.8 mg/dL) showed a significant decrease in blood flow-dependent vasodilation response (FMD) compared to the other group with hypouricemia (sUA > 0.8 mg/dL) [20]. Patients with functionally null mutations of URAT1 showed significantly impaired FMD. These findings suggest an association between RHU and impaired endothelial function (Figure 4). Tabara et al. reported that subjects with a nonsense mutation of URAT1 showed lower UA levels and reduced renal function. Taken together, RHU might be a risk factor for reduced renal function caused by impaired endothelial function. It should be noted that patients with RHU occasionally develop complications such as exercise-induced acute renal disability (EIAKI) [13]. EIAKI is thought to be caused by the constriction of the renal artery after exercise [13]. The prevalence of EIAKI is as high as 6% in patients with RHU, suggesting that increased ROS in RHU patients due to the reduced antioxidant action of UA impairs the endothelial function of the renal artery and induces its constriction via the reduced activity of protein kinase G in vascular smooth muscle cells (Figure 4). In addition to the impaired antioxidant action under extreme low SUA conditions, the higher oxidative stress level may be responsible for the endothelial dysfunction in hypouricemic patients. What is the cause of oxidative stress that is responsible for the impaired endothelial function in RHU? Xanthinuria is a hereditary disease characterized by the loss of function of XOR. We recently reported xanthinuria associated with a novel homozygous mutation (C1585T) of XOR [67]. While serum and urinary xanthine concentrations increased in this case, the SUA concentration was extremely low (0.1 mg/dL or less) [67]. Interestingly, despite the extremely low SUA concentration, the value of FMD was within the normal range, suggesting that ROS derived from XOR might be the source of ROS in hypouricemia (Figure 4). These results suggest that the suppression of XOR may be beneficial for impaired endothelial function in RHU, and this has been proven by the report showing that an XOR inhibitor improved the vasoconstriction of the renal artery and prevented the recurrence of EIAKI in patients with RHU [68,69]. After strenuous exercise, the elevated urinary UA concentration might obstruct the renal tubules in RHU patients, causing EIAKI. In addition to the increased excessive urinary urate excretion induced by exercise, an inflammatory signal via the NLRP3 inflammasome might be a mechanism of EIAKI in a mouse model using a high HPRT activity Urat1-Uox double knockout mouse [70].

Transient hypouricemia might impair endothelial function in the absence of NO. Becker et al. examined whether short-term hypouricemia induces the impairment of endothelial function under the condition of inhibiting NO production [71]. Seventeen young healthy men were enrolled in a randomized, double-blind, placebo-controlled three-way crossover trial. Under the inhibition of NO production by L-NAME (inhibitor of NO synthetase), the combination therapy of febuxostat and uricase (SUA level: 0.3 mg/dL) impaired heat-induced endothelium-dependent vasodilation compared to the placebo group (SUA level: 5.4 mg/dL) or the febuxostat alone group (SUA value: 2.1 mg/dL). In the febuxostat and uricase groups, an increase in lipid peroxidation was observed, which may reflect a decrease in the antioxidant capacity of plasma. These results suggest that extremely low levels of SUA could impair vasodilation mediated by endothelium-derived highly polarized factors (EDHF) in the absence of NO. EDHF is formed through classical or non-classical pathways. The classical pathway is activated by hyperpolarization factors released from endothelial cells. They hyperpolarize the plasma membranes of smooth muscle cells, causing a decrease in intracellular Ca^2+^ and the activation of K^+^ channels and relaxation. The non-classical pathway is activated by epoxyeicosatrienoic acid (EET) derived from endothelial cells, which activates K^+^ channels or transporters of smooth muscle cells to induce the hyperpolarization of the smooth muscle cells, in addition to their relaxation. Although SUA inhibits epoxy hydrolase, epoxide hydrolase inactivates EET under extremely low SUA conditions, resulting in the inhibition of heat-induced endothelium-dependent vasodilation [71] (Figure 4). These results suggest that secondary hypouricemia could deteriorate the function of EDHF and impair endothelial function, which might be associated with the loss of kidney function.

Secondary hypouricemia is defined as transient hypouricemia related to malnutrition, operation, or disease. Secondary hypouricemia is frequently observed in hospitalized patients. Hypouricemia in hospitalized patients is mostly transient [72]. Previous studies reported that the prevalence of hypouricemia in hospitalized patients was between 1.24% and 4.14%, with no gender difference (male: 2.26%, female: 2.93%) [44,73], which is higher than that in outpatients and in the general population. We recently studied the pathophysiology of hospitalized patients and found that the prevalence of hypouricemia was 11.1% (Table 2). Transient hypouricemia was predominant (93%) over persistent hypouricemia, suggesting that hypouricemia in hospitalized patients is secondary hypouricemia. We found that either emaciation, DM, malignancy, lymphoma, leukemia/MDS, SIADH, or medications including uric acid-lowering agents, chemotherapy, and TMP-SMX were significantly associated with hypouricemia patients compared to those of normouricea, which is supported by several studies [74].

It has been suggested that secondary hypouricemia or low levels of SUA might be a risk for death in hospitalized patients [75]. Although the mechanism of the hypouricemia-related death remains to be elucidated, it might be caused by the impaired endothelial dysfunction due to either degradation of NO or the impaired function of EDHF. This impaired endothelial function may play a pivotal role in the loss of kidney function, causing a risk of death in hospitalized patients. Further studies are required to address this question.

## 4. Treatment of Hyper- and Hypo Uricemia and Summary

Considering the different mechanisms of impaired kidney function in hyper- and hypo-uricemia, the methods of protection of kidney function differ between hyperuricemia and hypouricemia. To prevent hyperuricemia-induced kidney disease, urate lowering agents (ULAs) such as XOR inhibitors might be beneficial. The effects of XOR inhibitors on eGFR were examined [76,77,78,79,80]. A systematic review involving patients in the ULA intervention group and subjects in the control group, reported an improvement of eGFR and a reduction of the frequency of end-stage renal failure in the ULA intervention group [1]. Recently, in a CKD3 study, the XOR inhibitor topiroxostat reduced urinary albumin levels in asymptomatic hyperuricemic patients with Stage 3 CKD when the SUA were reduced below 6 mg/dL by topiroxostat [79]. In the FREED study [81], febuxostat improved renal events when SUA levels were reduced below 6 mg/dL. Thus, XOR inhibitors could improve renal function, and suppressed the onset of end-stage renal failure in CKD patients [77,78,79,80,81,82,83,84]. The JGMHG (Japanese Guideline for Management of Hyperuricemia and Gout) 3rd edition recommended that in order to protect kidney function in hyperuricemic patients with CKD, lowing SUA below 6 mg/dL by ULAs is conditionally recommended. Although recent reports, such as the PERL [85] and FIX-CKD [86] studies indicated that the xanthine oxidase inhibitor, allopurinol, failed to improve the kidney function in normouricemic patients with type 1 diabetes, and hyperuricemia patients with CKD, respectively, there are limitations in such a futile recruitment rate in FIX-CKD study and the rate of the missing of a primary outcome in the PERL study, respectively [87]. An additional randomized control trial may be necessary.

A treatment to protect kidney function in hypouricemia remains to be elucidated. Because of the pathophysiology of EIAKI in RHU, sufficient hydration, together with a vasodilator or dopamine as a Ca^2+^ antagonist is recommended. For its protection, the avoidance of anaerobic exercise is also recommended. The oxidative stress together with the precipitation of urate in tubules might play a pivotal role with regard to EIAKI in RHU; thus, treatment with an antioxidant XOR inhibitor could be beneficial. It has been reported that an XOR inhibitor, allopurinol, may be useful for protection from EIAKI in RHU patients [68]. The rationale for the use of allopurinol in this hypouricemic patient was to decrease the generation of UA, thus decreasing the filtered UA load and lowering the risk of precipitation of UA in the tubules, in addition to reducing oxidative stress. With this approach, the recurrence of renal injury was successfully prevented in a patient with RHU. The patient was able to resume their athletic lifestyle without repeat symptoms or recurrent AKI.

In conclusion, both hyperuricemia and hypouricemia cause urolithiasis, renal tubular damage, and kidney injury. With regard to the cardiovascular events, there is likely a U-shaped association of SUA with the loss of kidney function. Its underlying mechanisms could be attributable to two types of impaired endothelial function. The first is the endothelial dysfunction induced by intracellular UA, MSU and XOR under hyperuricemic conditions, while the other involves endothelial dysfunction induced by the depletion of NO and EDHF under hypouricemic conditions. To treat kidney disease in hyperuricemic patients, ULAs are recommended. To treat kidney disease in hypouricemic patients, XOR inhibitors might be useful.

## Figures and Tables

**Figure 1 biomedicines-11-01258-f001:**
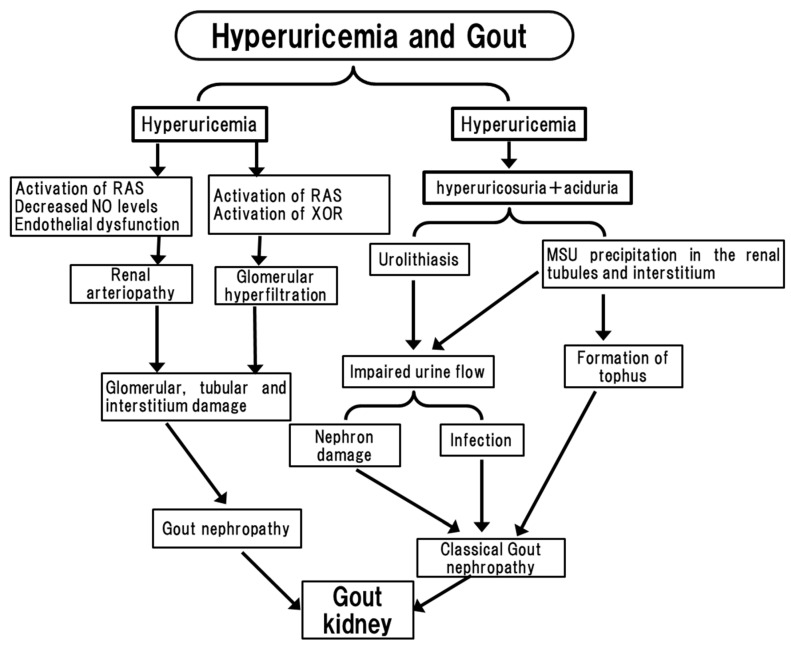
Pathophysiology of gout kidney. Hyperuricemia leads to hyperuricosuria-associated aciduria, forming urolithiasis and MSU precipitation. Hyperuricemia also influences several signals to exert renal arteriopathy and glomerular hyperfiltration, causing glomerular, tubular and interstitum damage. RAS: renin angiotensin system, NO: nitric oxide, XOR: xanthine oxidoreductase, MSU: monosodium urate.

**Figure 2 biomedicines-11-01258-f002:**
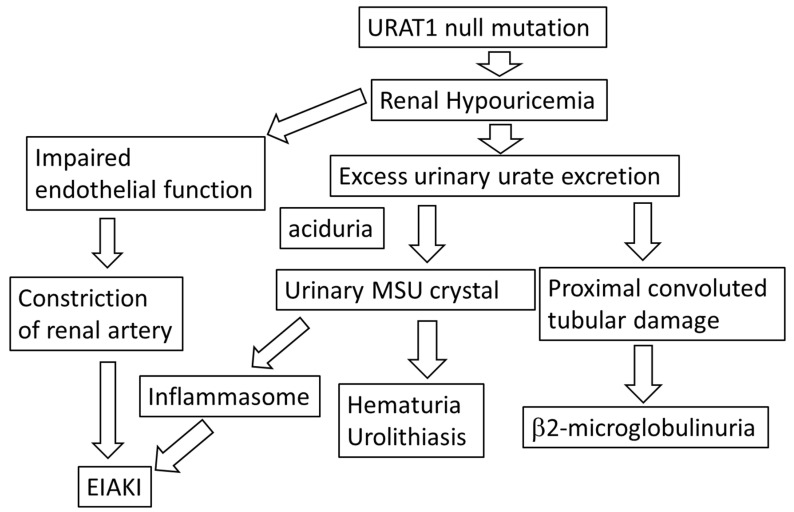
Kidney disfunction of RHU. A URAT1 mutation causes RHU. Excess urate excretion causes urinary MSU crystals, resulting in hematuria and urolithiasis. It causes proximal convoluted tubular damage and the elevation of urinary b2-microglobulin. Urinary MSU crystal activates the NLRP3 inflammasome, causing EIAKI. The loss of UA reduces the antioxidant action of the endothelial cells, causing the constriction of the renal artery and EIAKI. URAT1: uric acid transporter 1, MSU: monosodium urate, EIAKI: exercise-induced acute kidney injury.

**Figure 3 biomedicines-11-01258-f003:**
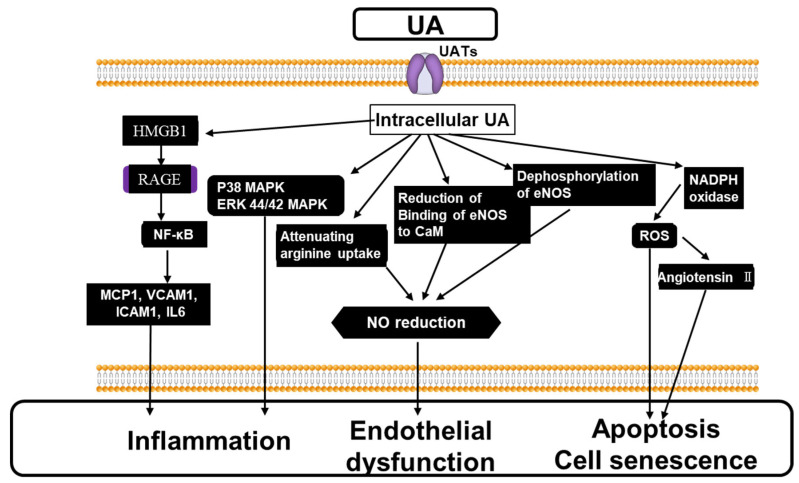
The mechanism of impaired vasodilation by the elevated extracellular and the resulting intracellular UA. The intracellular UA, through UATs, activates p38 44/42 MAPK, NF-kB, causing inflammation. The intracellular UA reduces NO levels via attenuating arginine uptake, the reduction of binding of eNOS to CaM, and the dephosphorylation of eNOS, leading to impaired endothelial function. The intracellular UA generates superoxide via the activation of NADPH oxidase associated with the production of Angiotensin II, causing apoptosis and cell senescence. See the text for details. CaM: calmodulin; NF-κB: Nuclear Factor-κB, MCP-1: monocyte chemotaxis protein-1, VCAM: vascular cell adhesion molecule, ICAM: intracellular cell adhesion molecule, IL6: interleukin 6, HMGB: High Mobility Group Box, eNOS: endothelial nitric oxide synthesis; NADPH: nicotinamide adenine dinucleotide phosphate; UA: uric acid; UAT: uric acid transporter.

**Figure 4 biomedicines-11-01258-f004:**
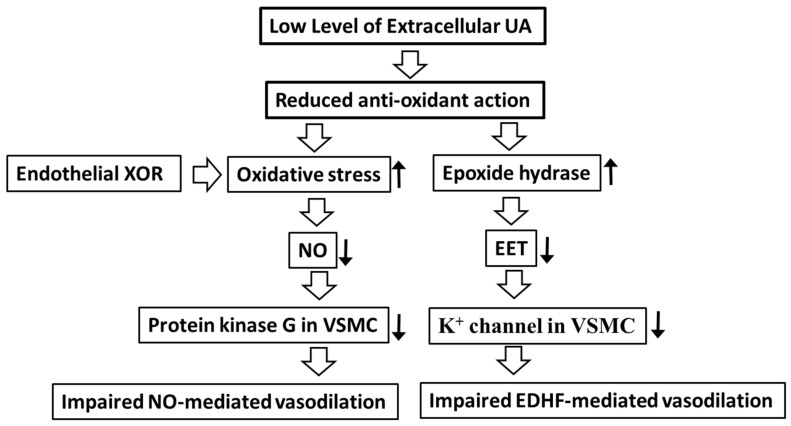
The mechanism of vasodilation promoted by the extracellular UA. The extracellular UA promotes vasodilation via the inhibition of oxidative stress and epoxide hydrolase, which indicates that the low serum UA leads to the impaired vasodilation. See the text for details. EDHF: endothelium-derived highly polarized factors; EET: epoxyeicosatrienoic acids; NO: nitric oxide; UA: uric acid; VCMC: vascular smooth muscle cell.

**Table 1 biomedicines-11-01258-t001:** The prevalence of hypouricemia in healthy subjects and outpatients.

Reports	Subjects	Total	Male	Female	Transient	Persistent
Wakasugi et al. (2015) [14]	Health checkup		0.21% (n = 90,710)	0.39% (n = 136,935)		
Tabe et al. (2015) [18]	Medical checkup		0.14% (n = 17,603)	0.4% (n = 3544)		
Kaneko et al. (1995) [19]	Medical checkup		0.09~0.12%	0.36~0.51%		
Matsuo et al. (2008) [15]	Self-defense forces	0.18% (n = 21,260)				
Hisatome et al. (1989) [17]	Outpatients	0.4% (n = 3258)			0.25%	0.15%

**Table 2 biomedicines-11-01258-t002:** Prevalence of hypouricemia in hospitalized patients.

Reports	Objects	Total	Male	Female	Transient	Persistent	Unclassified
Ramsdel et al. [72]	Admitted patients	0.97% (n = 6629)					
Ogino et al. [73]	Admitted patients	2.54% (n = 1220)	2.26% (n = 708)	2.93% (n = 512)			
Matsumoto et al. [74]	Admitted patients	11.1% (n = 675)	11.5% (n = 419)	10.5% (n = 256)	3.3% (22/675)	1.0% (7/675)	6.8% (46/675)

## Data Availability

Not applicable.

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
