# Peer review of "Impact of Hyper- and Hypo-Uricemia on Kidney Function"

_biomedicines, 2023, doi:10.3390/biomedicines11051258_

Round 1

Reviewer 1 Report

Dear Editor and authors 

This is an interesting review of how hyperuricemia and hypouricemia may induce renal injury.

1. Major issue

Before start reading, the sections need more simplified sructure with some highlights.

The 1st to 12th points are too troublesome for understanding the structure of this manuscript.

I would suggest the authors may regroup their points as following: 

Merge 1.2.3   into Section I: Introduction to hyperuricemia and hypouricemia: epidemiology and genetic mutation

Merge 4,5,6,7 into Section II: Clinical evidence of hyper- and hypo uremicemia may cause adverse effect on kidney (

Merge 8,9,10 into Section III: Molecular mechanism of hyper-, hypo, transient hypo- uremicemia may cause endothelial dysfunction

Merge 11,12 into Section IV: Treatment of hyper- and hypo uremicemia and summary

2. Minor issue:

a. figure1 : 

   the right <hyperuricemia> should be <hypouricemia>?

   The lower part of the figure: what is the difference between <gout nephropathy >, <classical gout nephropathy>, and <gout kidney>?

b. point 6:

   More explanation of <renal volume to resistive index> may be needed and why is it important?

c. point 7:

   U shape figure may be cited for prove the association of SUA and kidney disease.

d. point 10:

   Is secondary hypouricemia related to malnutrition? admission patients may suffered from operation or disease.

   How to define <secondary hypouricemia>?

e. All the figures should be generated by authors in this review or permission from origin should be gained.

   English edit and polishing is needed for concise this review.

f. Words in the Tables may not be in the bold font. The citation of report with the year published may be needed.

   Please check with the editor about the format of tables in MDPI.

Reviewer 2 Report

The review by Miake et al. delves into the epidemiology and genetic makeup of hyperuricemia and hypouricemia, along with the possible complications that may arise, such as urolithiasis, renal tubular damage, and renal problems in patients with dysuricemia, such as hyperuricemia, gout, and hypouricemia. In addition, the U-shaped correlation of AUB with renal failure is explored, and a small analysis of possible treatments is performed. The paper is well-written and clear, and the references are well-used. It is suggested to accept the article after the following minor corrections.

Line 23: Change " suppresses" to " suppresses". The singular verb “suppresses” does not appear to agree with the plural subject “the most potent antioxidants”. Consider changing the verb form for subject-verb agreement.

Line 57: Define CKD

Line 57: Change "Eepidemiological " to "Epidemiological ".

Line 72: Change "epidemiology and…" to "the epidemiology and…".

Tables:  References are incomplete. Place the references in the same format as the text. Capitalize the first word of the row headers.

Figure 1:

-       What is the difference between the hyperuricemia boxes at the top? Be more specific, or put it together in the same box.

-       Define the abbreviations in the figure caption:  RAS, XOR, NO, MSU.

-       The title does not adequately represent the illustration. The title seems to show the relationship between hyperuricemia and the generation of gout, according to the orientation of the arrows. Therefore, the title would have to be modified.

Figure 2:

-       Define the following abbreviations in the figure caption: URAT1, MSU, EIAK

Figure 3:

-       Define the abbreviations in the figure caption: ROS, CaM, Nf-κB, MCP-1, VCAM-1, ICAM, IL6, HMGB.

Figures:

-       Homogenize the style of the arrows and boxes of all the images.

Line 181: Define “MSU” the first time it is used.

Line 202: Define “CKD” the first time it is used.

Line 213: Change "renal artery " to "the renal artery ". The noun phrase renal artery seems to be missing a determiner before it. Consider adding an article.

Line 442: A conclusion is missing. Is the summary section the conclusion? Change the title to "final comments" or "conclusions"

Line 250: Change " been also" to "also been". The adverbial also appears to be misplaced in this sentence. Determine the appropriate placement for the adverb.

Round 2

Reviewer 1 Report

I have no further questions, thanks.

However, english editing may still be needed to concise the review.

Author Response

Dear Reviewers

I am sending a revised manuscript (biomedicines-2320604) entitled " Impact of Hyper- and Hypo-uricemia on Kidney Function" by Miake J, et al. Our answers to each point raised by the reviewers are also provided. The comments of the reviewers were very helpful in improving the presentation, and we appreciate them indeed. All the points raised by the reviewers have been carefully checked and incorporated into the revised manuscript. The reasons for each revision are described as follows.

We hope that the revised version will be suitable for publication in Biomedicine.

Reviewer 1

  1. Minor issue

However, English editing may still be needed to concise the review.

Thank you for your comments. Regarding the English editing, we would like to leave it to the production team of the journal, since editorial office suggest that English editing will be performed by production team of the journal team .